# Impact of the Filling Medium on the Colour and Sensory Characteristics of Canned European Eels (*Anguilla anguilla* L.)

**DOI:** 10.3390/foods11081115

**Published:** 2022-04-13

**Authors:** Lucía Gómez-Limia, Javier Carballo, Miriam Rodríguez-González, Sidonia Martínez

**Affiliations:** Food Technology, Faculty of Science, Campus As Lagoas s/n, University of Vigo, 32004 Ourense, Spain; lugomez@uvigo.es (L.G.-L.); carbatec@uvigo.es (J.C.); miriamrodrigzg@gmail.com (M.R.-G.)

**Keywords:** olive oil, sunflower oil, spicy olive oil, canned eel, colour, sensory analysis

## Abstract

The different vegetable oils used in canned fish as a filling medium have a preserving effect and contribute to the palatability of the product. In this study, the colour of European eels and the filling medium (sunflower oil, olive oil or spicy olive oil) was measured at different steps of the canning process. The sensorial characteristics of canned eels packed in the different oils were also evaluated. Colour scores (CieLab values) were higher in canned eels packed in sunflower and spicy olive oil than in canned eels packed in olive oil. The changes in colour parameters depended on the type of oil, the stage of the process and the storage time. Colour changes in canned eels packed in olive oil were highest during the sterilization process. Spicy olive oil was the filling medium in which the colour change was greatest, probably due to the migration of some of the spice components into the oil. Organoleptic properties were directly related to the type of oil used as the filling medium. The canned eels packed in sunflower oil were those awarded the highest scores in consumer tests, although the preferences varied depending on the age and gender of the consumers.

## 1. Introduction

Sensory properties are one of the most important quality attributes of any product and directly determines consumer satisfaction or overall acceptance based on colour, taste, flavour and aroma. The level of acceptance by consumers plays a key role in the development of new food products. Consumer decisions about food products largely depend on their sensory profile [1].

The colour of food has an important influence on consumer opinion about the quality of the product. Colour provides one of the most important visual cues regarding the sensory properties (e.g., taste and flavour) [2]. Colour is correlated with other quality parameters such as sensory and nutritional characteristics, and it can help to indicate defects [3,4]. It can also be used as an indicator of the intensity of heat treatments and to predict the deterioration resulting from heat exposure [5]. Colour can be determined by visual inspection or through instrumental assessment. Visual inspection is rather subjective and varies greatly from one person to another. Robust results can only be obtained by using trained assessors. As the visual inspection method is time consuming and involves specially trained observers, the instrumental method is recommended for conducting colour analysis. Colour spaces and numerical values are usually used to construct models to represent colours in two- or three-dimensional spaces [4,6].

The sensory qualities of food are very important in determining whether consumers will accept or reject the product. The sensory quality of fish or a fish product can be established by evaluation of various sensory attributes such as appearance, colour, taste, texture, aroma and flavour. However, these attributes can undergo changes during processing.

Canning is one of the oldest methods of food processing. Canned fish are economically important products in many countries, supporting a high market demand due to convenience and affordability. Canned fish have a relatively long shelf life and are thus widely available at all times. The synergistic effects of heat treatment and the filling medium plays an important role in the canning process. The procedures used to prepare raw fish for canning, the pre-cooking process, type of heat treatment, filling medium and storage conditions are carefully selected to lengthen the shelf life, preserve the quality and achieve optimal sensory characteristics. The quality of canned fish can vary with different factors such as the quality of the raw product, processing conditions, packaging material and filling ingredients [7]. The sensory and nutritional characteristics are the result of complex interactions between the fish and filling medium. Colour and sensory properties are among the most important parameters for determining changes in food quality, and they have an important impact on the acceptability of canned fish. In the canning process, brine or oil are generally used to improve the taste, aroma and appearance of the fish and to yield a compact product with a soft, juicy texture. The vegetable oils in which the fish are packed have a preserving effect and contribute to the palatability of the product. On the other hand, the oils can be used as part of a commercial strategy of adding value to canned fish. The filling oil can dilute and/or partly extract some components, and lead to heat transfer in fish muscle [8]. Processing can also cause changes in the sensory and physical properties of the oils. Olive and sunflower oils are two of the most common filling media used in canning.

Eels are commercially valuable in Europe (mainly Spain, Portugal, Italy and the Netherlands) and Asia (mainly Japan, China, Korea and Taiwan). The eels can be processed in different ways such as freezing, smoking, canning and jelling. Although canned eel is a commercially important product in some countries, information on its quality parameters is scarce. Eel populations have declined greatly because of various factors threatening their survival, such as environmental impacts associated with the construction of diverse obstacles in rivers and an increase in pollution of human origin [9]. Canning enables eels to be consumed throughout the year, while respecting restrictions or quotas aimed at protecting the species. Larger eels are least valued for fresh consumption, and their exploitation, instead of smaller eels (the commercial weight is usually between 150 and 250 g, depending on the country of sale), could help to increase the reproductive success of the species.

Although different filling media are widely used for canning fish, available information on how different oils affect the sensory quality of fish is scarce, and the effects on the sensory properties of canned eels have not been investigated.

In this context, the aim of the present study was to evaluate how the colour of European eels (*Anguilla anguilla* L.) and different filling media (vegetable oils) were affected at different stages of the canning process by the type of oil used and by the storage time. The sensorial characteristics of canned eels were also evaluated.

## 2. Materials and Methods

### 2.1. Samples and Fish Preparation

#### 2.1.1. Samples Preparation

Eels (weighing between 200 and 400 g) were purchased from a local market (Mariscos Vivos del Grove, Plaza de Abastos,) in Ourense (Galicia, NW Spain) after being captured in the Ulla River (Galicia, NW Spain). The eels were eviscerated and transferred to the laboratory.

The fish was washed, packed in vacuum bags and stored at −20 °C until further processing.

#### 2.1.2. Eel Canning Process

Before canning, frozen eels were thawed in 12% brine. The head and skin were removed, and the fish were sliced. The fish slices were fried at 190 °C for 2 min in a conventional frying pan to eliminate the water present and to prevent the formation of a water-oil mixture in the cans. Two different frying media were used: refined sunflower and an olive oil mixture (refined olive oil + virgin olive oil). The fried eel slices were cooled (30 °C) and placed in glass jars (6 or 7 slices in each). The hot filling medium was then added: refined sunflower oil (eel slices previously fried in this oil), olive oil (refined olive oil + virgin olive oil) or spiced olive oil (refined olive oil + virgin olive oil plus chilli and pepper) (eel slices previously fried in olive oil) (Figure 1). In the case of the spiced olive oil, approximately 0.20 g of dried chilli and 0.30 g of dried pepper were added inside the glass jar before sterilization.

The jars were vacuum-sealed and sterilized at 118 °C for 30 min, after which they were cooled and stored at room temperature in a dark room.

For colour determination, the eels and the different oils were sampled at different stages: before processing, after frying the fish and after the sterilization treatment.

#### 2.1.3. Storage

The canned eels and oils were also sampled after two months and one year of storage at room temperature. The jars were opened, and the filling medium was carefully drained off through a 3 mm pore sieve. The sensory analysis of the canned eels was carried out after one year of storage.

### 2.2. Colour Determination

Colour measurements were made with a portable colorimeter (Chroma Meter CR−400, Konica Minolta Sensing, Inc., Osaka, Japan). The colour was expressed by the values of the coordinates in the CIEL**a*b** space: *L** (lightness), *a** (balance between red and green) and *b** (balance between yellow and blue).

The colorimetric parameters hue angle (*H*°) (colour of sample as defined by its location in a 360° axis; 0 or 360° = red, 90° = yellow, 180° = green and 270° = blue), chroma (*C**) (colour saturation, increasing from 0) and total colour change (Δ*E*) were calculated as follows:*H*° = 180° + arctg (*b**/*a**), for *a** < 0 and *b** > 0


$$C*=\sqrt{{\left(a\right)}^{2}+{\left(b\right)}^{2}}$$


Δ*E* = √[(*L*_0_ − *L*)^2^ + (*a*_0_ − *a*)^2^ + (*b*_0_ − *b*)^2^], colorimetric coordinates of raw eels indicated by “0”.

All determinations were made at least in triplicate.

### 2.3. Sensorial Analyses

#### 2.3.1. Descriptive Sensory Evaluation

The aim of the sensorial analysis was to identify the important sensory attributes of canned eels in different filling media. A panel of 11 trained tastes recruited from the Faculty of Science of Ourense (University of Vigo, Spain) was used to evaluate the intensity of each attribute.

The panellists were trained in sensory evaluation of fish according to International Organization for Standardization (ISO) standards [10] and they were familiarized with the questionnaire on the sensory parameters.

The different descriptive attributes were measured according to ISO [11,12] standards. The final descriptions selected were defined according to UNE-EN-ISO 5492:2010/A1:2017 [13]. Sensory descriptive terms (15 attributes) were generated and agreed on among the panel members. The descriptive attributes and their definitions are included in Table 1. The descriptive parameters of fish (bitterness, acid, sweet, salty, metallic, pungent, colour, gloss, appearance, aroma intensity, preference, hardness, adhesion, residual taste in the mouth and aftertaste) were evaluated by the panellists on a scale ranging from 1 (absence of sensation) to 10 (extremely intense sensation) (Figure 2). The sensory profile of the samples was determined from the scores awarded in the descriptive analysis.

A colour scale was created to enable the colour of the samples to be evaluated, taking into account the different shades that the fish acquired during treatment (Figure 3).

#### 2.3.2. Consumer Tests

Samples of canned eels packed in the different oils were also used for the consumer test. The test consumer group was formed by 97 volunteers from the teaching staff, graduate students and technicians of Faculty of Science (Ourense, Spain). The testers were of both sexes (63 women and 34 men) and different ages (31 < 20 years; 46 from 21 to 35 years; 20 > 35 years). Prior to consumer testing, a supplementary questionnaire was filled out to obtain more information about the age and gender of the testers and their weekly fish consumption. The panellists ate fish products often and they were notified about the fish that they had to taste.

The consumer study involved analysis to investigate the perceptions of consumers on the previously selected attribute group, and degree of acceptance. The testers were divided into two groups, with 5–6 individuals in each session. The testers evaluated three samples identified with a three-digit code, corresponding to the three different types of canned eels (packing in sunflower oil, olive oil or spiced olive oil) in each session. Testers ate pieces of bread and sipped water to neutralize the taste between sample testing. The samples were presented simultaneously to testers.

The parameters of flavour, aroma, texture, appearance and general appreciation were evaluated by the testers in a 10-point hedonic scale test, with a score of 1 indicating extreme dislike, and a score of 10 indicating excellent (Figure 4). The final scores reported were the average scores for the 97 testers.

### 2.4. Statistical Analysis

All analyses were conducted at least in triplicate. The data were examined by analysis of variance (ANOVA). The least significant difference (LSD) test was applied, with a 95% confidence interval (*p* ≤ 0.05), for comparison of the mean values, using the statistical software Statistica version 8.1 (Statsoft© Inc., Tulsa, OK, USA).

The sensory data were examined by Friedman analysis of variance, to test for any differences between the overall preferences for canned eels. The significance of differences between samples was determined by the Fisher test (α = 0.05), modified for non-parametric data.

## 3. Results and Discussion

### 3.1. Colour

Vision is usually the first sense used in making purchasing decisions. The colour of food products is a very important quality parameter for consumers.

The colour parameters measured in raw eels and in canned eels sampled at each stage of the canning process (frying, sterilization, and after room storage for 2 and 12 months), and packed in sunflower oil, olive oil or spiced olive oil (olive oil plus chilli and pepper) are shown in Table 2.

The *L** coordinate represents the luminosity and varies between 0 (black) and 100 (white). The raw eel presented a mean luminosity value of 74.05. This is higher than values reported for other species. Huang et al. [14] reported an *L** value of 45 in Pacific tuna. Fuentes-Zaragoza et al. [15] reported a value of 47.96 in hake, and Sánchez-Zapata et al. [16] reported a value of 57.47 for this same species. *L** values are generally lower in fatty fish with high proportions of dark muscle [17]. Greater luminosity is usually related to greater whiteness, a parameter that is valued in some fish such as hake or cod. Lower luminosity is usually associated with older and/or stale products. In fish with low concentration of pigments, such as carotenoids or haemopigments, reflection of light is favoured, and the *L** value will therefore be higher.

The *a** coordinate represents the variation between red (>0) and green (<0). High *a** values are usually related to a higher concentration of haemopigments, mainly myoglobin, in the dark muscle and lower concentrations of carotenoids [18]. In the raw eel muscle, the low *a** values (mean −4.95) indicated a light colouration, with slightly greenish tones.

The *b** coordinate represents the variation between yellow (>0) and blue (<0). In the raw eels the mean *b** value of 14.60 indicated slightly yellowish tones. Variations in this value are usually due to the presence of a greater or lower amounts of red muscle [16].

Hue angle (*H*°) is used to describe the difference of a certain colour with reference to grey colour with the same lightness. It varies between 0 and 360°. In this case, 90° represents yellow, 180° represents blue-green and 270° represents blue [19]. In the raw eel, the mean *H*° value of 108.74° indicated yellowish-greenish tones.

The *C** coordinate, or chroma, represents the purity of the colour, and is used to define the degree of difference of a hue in comparison to a grey colour with the same lightness. The *C** mean value in raw eels was 15.15. In previous studies, it has been pointed out that the chroma parameter generally performs in a similar way to *a** and *b**, i.e., the factors that modify these coordinates also affect the *C** values [20].

In food processing, colour serves as a cue for temperature and time control during cooking/processing and is correlated with changes in aroma and flavour.

Frying caused a decrease in brightness (*L**) relative to raw eels. This decrease may be due to chemical reactions that take place in the frying oil, such as hydrolysis, oxidation, polymerization and others [21].

Frying did not cause any significant changes in parameter *a**. However, the *b** and *C** values increased. This indicates a tendency towards yellowish colours. The *b** and *C** values were higher in the eels fried in olive oil than in eels fried in sunflower oil. Frying caused a decrease in hue angle (*H*°) in eels fried in olive oil, but not in those fried in sunflower oil. During frying, heat and mass transfer occurs between the fish and the oil, causing physicochemical changes that affect the colour of the fish and the oil. Caramelization and Maillard reactions also take place at the surface, and evaporation of water and absorption of oil modify the colour of the fried product. Frying often results in darkening of the muscle due to these reactions. The colour changes during frying are determined by the temperature of the oil, the type of oil and the shape and size of the food [22]. Pérez-Palacios et al. [23] reported more intense browning in hake fillets fried in olive oil than in those fried in sunflower oil.

The sterilization treatment caused a significant increase in the luminosity of the canned eels packed in sunflower oil and olive oil, but not in canned eels packed in spicy olive oil. The *a** coordinate did not vary significantly during the sterilization process, but the *b** and chroma values increased in the eel packed in sunflower oil. Hue angle decreased during sterilization in eels packed in sunflower oil. In the spiced eels, *b**, chroma and hue angle did not vary during sterilization. The denaturation of myoglobin and the oxidation of some pigments such as carotenoids cause changes in the colour of the muscle from red to paler pinkish colours, which leads to an increase in luminosity. The increase in *L** values could also be due to the leaching of fish pigments or of white connective tissue containing collagen between the segments of muscles [24]. In the canned eels packed in spicy olive oil, the spices appear to significantly affect the colour parameters, probably due to the presence of pigments or prevention or enhancement of certain reactions in the fish muscle.

In the canned eels packed in sunflower oil, the *L**, *b** and *C** values remained stable during the first two months of storage, but *a** approached positive values and *H°* decreased. After storage for 12 months, the *L** value decreased in these canned eels. In the canned eels packed in olive oil, the *L*, b** and *C** values decreased during the first two months of storage. In this case, *L** and *H°* continued to decrease until twelve months, and the value of *a**, *b** and *C** values increased. In canned eels packed in spicy olive oil, a decrease in *L** was observed after two months of storage. These changes can be explained by denaturation of some pigments and haem-proteins inside the fish muscle [25]. The changes in colour in fish during storage are also associated with oxidation of pigments and the decomposition and polymerization of primary products of lipid oxidation [26]. Changes in filling medium also affect the colour of the fish.

Canned eels packed in olive oil underwent the greatest global colour change (Δ*E*) during the sterilization process (mean values of 13.31). The changes in all the colour parameters are considered indicative of the modification of the colour in the fish and the filling medium because of the heat treatments applied and processes (such as oxidation) occurring during storage.

The colour of oils is influenced by the pigment content and has been widely used as a quality index. The colour parameters measured in the raw sunflower oil and fresh olive oil and in sunflower oil, olive oil and spicy olive oil after each step of the canning process and after storage for 2 and 12 months are shown in Table 3.

No significant differences were observed between raw sunflower oil and olive oil in regard to luminosity (*L**). In sunflower oil, the different steps during canning and storage did not cause significant changes (*p* > 0.05) in this parameter. However, in olive oil, the *L** value was significantly lower (*p* < 0.05) after the frying process. The decrease in *L** is associated with darkening of the oil, due to the formation of polymers and dissolution of non-polar compounds in the medium [20]. The loss of luminosity may also be associated with the formation of the Maillard products resulting from interactions between food and the frying oil [27]. Sterilization causes significant changes in *L** values, relative to the fresh oil, in olive and spiced olive oil. Storage for 2 months caused an increase in luminosity in olive oil. With regard to spicy olive oil, the luminosity increased during the first months of storage, but decreased after 12 months. These changes may be due to the exchange of substances between the food and filling oil. Sánchez-Gimeno et al. [20] observed an increase in luminosity of fish fried in olive oil, but a decrease in the same fish fried in sunflower oil.

Regarding the greenness/redness (*a** value), as expected the values were lower in olive oil (−10.49) than in sunflower oil (−3.85) due to the greenish hue of the former. In sunflower oil, the *a** values decreased after frying (−4.72) and after sterilization (−4.16). During storage, *a** initially remained stable (2 months), but then decreased after 12 months of storage. The *a** values in olive oil and spicy olive oil were higher (redder colour) after frying and sterilization and at the beginning of storage and decreased after 12 months of storage. Positive *a** values (*a** > 0, redness), which would indicate a less greenish colouration, were not observed in any of the samples. Reddish colouration is not acceptable from the point of view of oil quality as it is related to combined oxidized fatty acid and pyrolytic condensation products [28].

The *b** value was lower in sunflower oil (4.05) than in olive oil (27.46). The colour of oils is chiefly influenced by two groups of constituents: carotenoids and chlorophyll. A yellow colour is associated with the presence of carotenoids in oils, while a green colour is associated with the presence of chlorophylls. Carotenoids predominate in sunflower oil and are responsible for the yellow colour of the oil [29]. Olive oil contains more chlorophyll which gives the oil a greenish colour. In addition, the extraction technology and the maturity of the fruit from which the oil is extracted determine the intensity of the colour of the oil.

Regarding sunflower oil, an increase in the *b** value was observed after frying. Browning reactions, which are caused by oxidation products in the frying oil, can change the colour of the oil [27]. During the sterilization process and the first months of storage, the *b** value remained stable and increased after 12 months. The increase in *b** values may be due to the oxidation of carotenoids and to the exchange of different components between the eel and the filling medium. Sánchez-Gimeno et al. [20] reported that the *b** values increased in olive oil and sunflower oil after frying.

The *H*° values were higher in raw sunflower oil (133.50°) than in olive oil (110.95°). The frying treatment decreased this parameter in sunflower oil, but not in olive oil. There were no changes in *H*° values in olive oil or spicy olive oil after the sterilization process; however, an increase in this parameter was observed in sunflower oil after this treatment. After 12 months of storage, *H°* values decreased in the three types of filling media.

The changes in the *C** values (chroma) followed the same trend as in *b** values. The *C** value was lower in raw sunflower oil (5.59) than in raw olive oil (29.40). Frying produced an increase in *C** values in sunflower oil, while sterilization caused a decrease in these values. No significant changes were observed in the *C** values in olive oil after frying and sterilization. The *C** values increased after 12 months of storage.

Δ*E* represents the overall colour change. The greatest changes took place in the spicy olive oil and mainly during storage. The more pronounced changes in the spicy olive oil may be due to migration of some of the components of the pepper and/or chilli into the oil.

Vegetable oils are susceptible to oxidation. Oxidation is an important cause of deterioration in oil quality during processing and storage [30]. A significant difference (*p* < 0.05) in most colour parameters was observed in canned filling media relative to the control sample. The thermal stability of oils depends on their chemical structure. Reda [31] reported that saturated oils are more stable than unsaturated oils. Moreover, during storage, the colour of the oil changes substantially over time. These changes may be due to the dilution and partial extraction of some components [8]. Light, heat and oxygen cause the degradation of pigments such as chlorophylls and carotenoids that give the oil its colour. Chlorophylls and carotenoids play an essential role in the oxidative stability of oils [32]. The loss of colouration during heat treatments is related to the thermolability of the pigments present in the oils.

### 3.2. Sensorial Analysis

#### 3.2.1. Sensory Description

The sensory quality of a food is the result of the interaction between the product and consumer. Acceptance or rejection of food by humans is based on the sensations it produces. The sensory quality of a food is evaluated by sensory analysis. Sensory analysis is an organized assessment of the colour, aroma, flavour, texture and appearance of a food. Different studies on fish consumption reveal that the principal factors responsible for the acceptance or rejection of a product are sensory characteristics [33].

The differences observed in the descriptive tests, in relation to the intensity of each of the attributes, can be observed in Figure 5, which shows the different sensory profiles of the canned eels, according to the perceptions of the panellists.

Prior to consumption, the external aspect influences decisions on the intention of purchase or consumption of a food by affecting expectations of palatability [34]. Colour significantly affects food acceptance. Colour scales are available for evaluation of some fish, such as salmon and tuna, and for the skin of some aquarium fish. However, colour scales are seldomly used for fresh or processed muscle of other fish. In this work, a colour scale was developed with the different colourations observed in the eel muscle after different treatments (Figure 3). The colour parameters revealed differences between canned eels due to the colour of each of the oils used as filling medium. Colour scores were higher for canned eels packed in sunflower and spicy olive oil than for canned eels packed in olive oil. These findings are consistent with the CieLab values obtained (Table 2), while the values for canned eel packed in sunflower oil and spicy olive oil after 12 months of storage were more similar to each other than those of olive oil. The canned eels packed in olive oil were yellower-greener and canned eels packed in sunflower and spicy olive were pinker. The olive oil used as filling medium, due to its characteristic organoleptic imprint, could had partially attenuated the typical eel colour. Colour scores were significantly negatively correlated with *a** (r = −0.99), *b** (r = −0.98), *C** (r = −0.97) and Δ*E* values (r = −0.97) and positively correlated with *H*° (r = 0.97).

The highest scores for glossiness and appearance were also obtained for canned eels packed in spicy olive oil.

Texture is an important parameter of fish also associated with consumer acceptability [35]. The texture is strongly related to the amino acid composition and protein structure, which undergo important changes during processing and storage. However, hardness was lower in canned eels packed in olive oil than in canned eels packed in sunflower oil and spicy olive oil. This may be due to the different changes that proteins and amino acids undergo during the process of canning eels in different filling media [36]. No differences in aftertaste residual or adhesiveness were observed between the different canned eel products.

Aroma is an important sensory property of fish products, and it can provide information about any physical/chemical alterations that have occurred during handling and processing. Canned eels packed in spicy olive oil were characterized by lower scores for aroma intensity. Canned eels packed in sunflower oil were awarded higher scores for preferred aroma. The oils used as filling media provide characteristic aromas to the fish.

As expected, the panel of assessors perceived significant differences in the pungency of the canned eels packed in spicy olive oil and the other two canned eel products (Figure 5B). The pungent taste is due to the chilli and pepper contained in spicy canned eels. Spicy canned eels were also slightly more bitter and acidic. Capsaicinoids are responsible for the spiciness of chili and peppers and are directly related to the pungency. On the other hand, different studies have indicated that capsaicin can evoke a bitter taste [37]. Organic acids present in spices can also contribute to the acidity of the canned fish.

No differences in basic taste were observed between canned eels packed in sunflower oil and canned eels packed in olive oil.

These results revealed that the filling medium significantly affected the sensory properties of the canned eels.

The non-parametric Friedman’s test is usually used to analyse the data obtained from a multiple-samples ranking test [38] and allows the establishment of the degree of probability with which it can be shown that panellists recognize differences between the samples tested. The F value was calculated according to the standard, yielding a value of 13.27. Taking into account this value and the numbers of samples and panellists it is confirmed that the participants in the sensory analysis were able to detect differences between the canned eels, as the observed F value was higher than the estimated F value (5.99 for α = 0.05).

#### 3.2.2. Consumer Preference

The average acceptance value scores awarded by 97 testers, expressed in %, are shown in Figure 6. Participants responded on a scale between 1 (extreme dislike) and 10 (excellent). A high proportion of the testers rated all canned eels with positive expressions on the hedonic scale. The test product with the greatest number of positive evaluations was the canned eels packed in sunflower oil, followed by the spicy canned eels.

In the case of the purchase intention, 45% of the participants would buy the canned eels packed in sunflower oil, 24% would buy canned eels in spicy packed olive oil and 31% would buy canned eels packed in olive oil.

The sense of taste may play an important role in food preferences and food choices. In consumers, different factors, such as demographics and social factors, can affect the overall scores regarding satisfaction with products. Numerous studies have attempted to clarify the influence of gender and/or age on sensory perception of and preference for different foods [39,40]. Age, gender, employment status and frequency of consumption of canned fish or shellfish are very important factors. In this tasting, 76% of the participants were regular consumers (once or twice a week) of this type of product. The average female and male consumer preferences for canned eel attributes are illustrated in Figure 7A. The results indicated that females preferred spicy canned eels packed in olive oil (41%) and canned eels packed in sunflower oil (37%), while men mostly preferred canned eels packed in sunflower oil (65%). Previous studies of the effects of gender on sensory function indicated that females have more sensitive and more accurate sensory perception than males [41].

The age of the consumers varied as follows <20 years old (about 31.96%), between 21 and 35 (about 47.42%) and the remaining were over 35 years old (20.62%). It was found that 63% of tasters under 20 years of age preferred canned eels packed in sunflower oil, and only 20% preferred canned eels packed in olive oil. Those over 35 years of age also showed similar preferences (55% preferred canned eels packed in sunflower oil and 20% canned eels packed in olive oil). The middle group showed a similar preference for canned eels packed in sunflower oil and canned eels packed in spicy olive oil.

Mojet et al. [42] studied the effect of gender and age on the threshold sensitivity of the basic tastes in 22 young adults (11 males and 11 females) and 21 older adults (10 males and 11 females). They reported that age, but not gender, significant y affected the sensitivity.

The consumers generally considered that all of the canned eel products were acceptable, with moderately high scores awarded to all. However, canned eels packed in sunflower oil were ranked the best of the three canned eels in sensory acceptation.

The type of filling medium played an important role in the formation of volatile compounds during the preparation of canned fish. Some filling media can slightly alter the typical fish sensory characteristics.

## 4. Conclusions

Marked changes in the colour of canned eels and the filling medium were detected as a result of the canning process and subsequent storage. The changes that took place depended on the processing step, the filling medium and the storage time. Luminosity, *b** (variation between yellow and blue), hue angle and chroma were the parameters most affected. The addition of spices (pepper and chilli pepper) to the filling medium caused important changes in the colour of the medium. The changes in the spicy olive oil may be due to migration of some of the components of the pepper and/or chilli into the oil. The changes in the colour in the canned eels and filling medium during the manufacture of canned fish products have an important effect on the perceived quality.

A sensory analysis of canned eels was carried out after the cans had been stored at room temperature for one year. The canned eels packed in sunflower oil were awarded the highest scores by testers, who indicated that this was the product that they were most likely to purchase. Therefore, it is concluded that canning is a good option for larger eels, due to the high level of acceptance. However, the preferences varied depending on the age and gender of the consumers. Females preferred spicy canned eels packed in olive oil and canned eels packed in sunflower oil, while males mainly preferred canned eels packed in sunflower oil. The youngest and oldest consumers preferred the eels canned in sunflower oil and those of 21–35 years old equally preferred the eels canned in spicy olive oil and those canned in sunflower oil.

The present results on colour and sensory quality provide valuable information regarding consumer acceptance of canned fish and the effect on sensory properties that may be expected as a result of using different filling media in canned fish.

## Figures and Tables

**Figure 1 foods-11-01115-f001:**
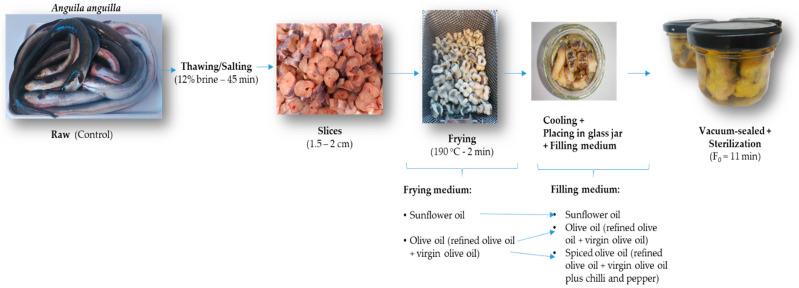
Flow chart of the canned eel production process.

**Figure 2 foods-11-01115-f002:**
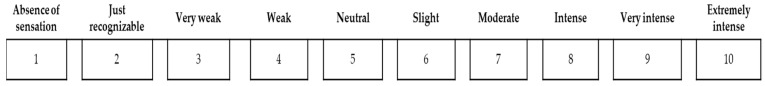
Scores of intensity of each descriptor.

**Figure 3 foods-11-01115-f003:**
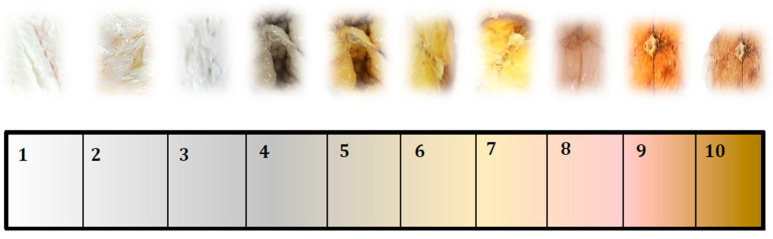
Colour scale prepared for rapid assessment of the colour of eel muscle (1= lighter and whitish to 10= darker and brownish).

**Figure 4 foods-11-01115-f004:**
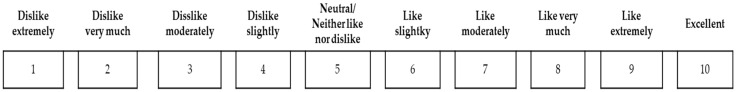
Hedonic 10-point scale.

**Figure 5 foods-11-01115-f005:**
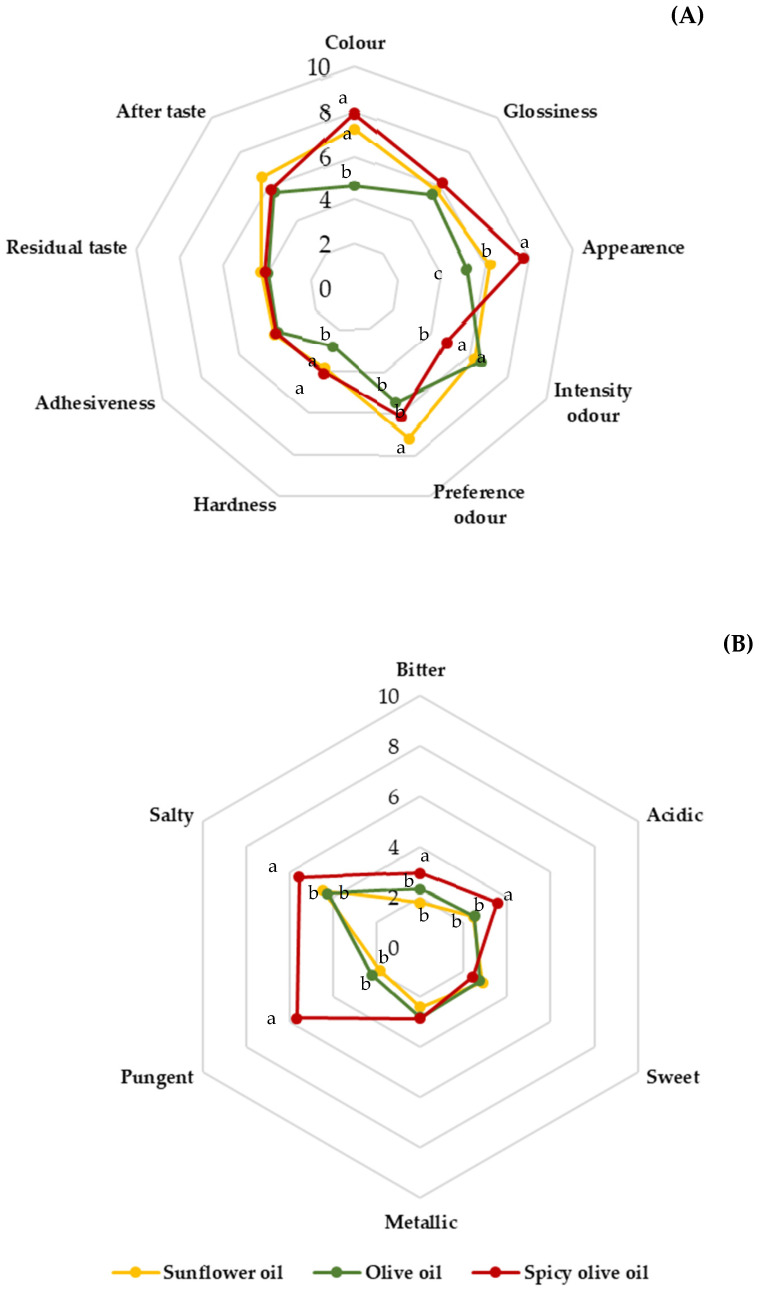
Graphic expression of the sensory profiles of canned eel packed with different filling media. (**A**): texture in mouth and aroma; (**B**): taste. Plotted values are the means of 11 observations. Different letters indicate significantly different (*p* < 0.05).

**Figure 6 foods-11-01115-f006:**
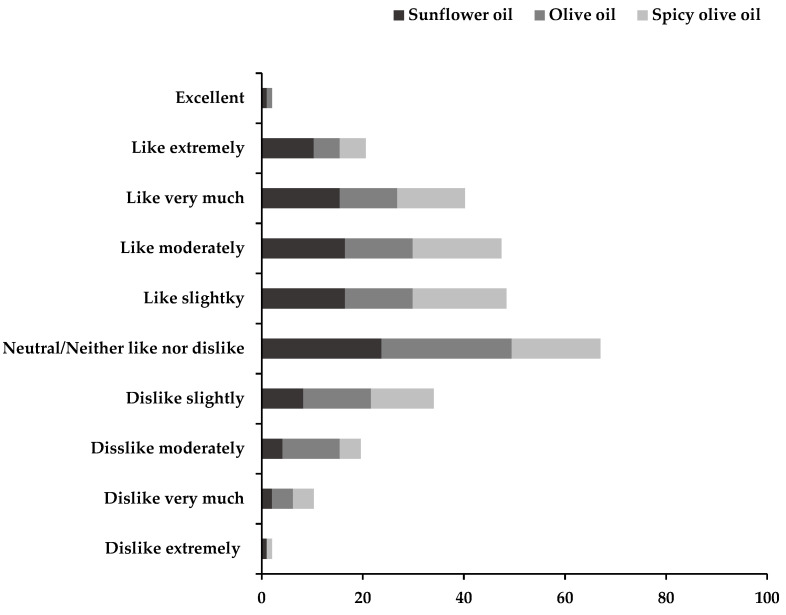
Distribution of consumer acceptance scores for three candy bars. Average acceptance values scores awarded by 97 testers expressed in %.

**Figure 7 foods-11-01115-f007:**
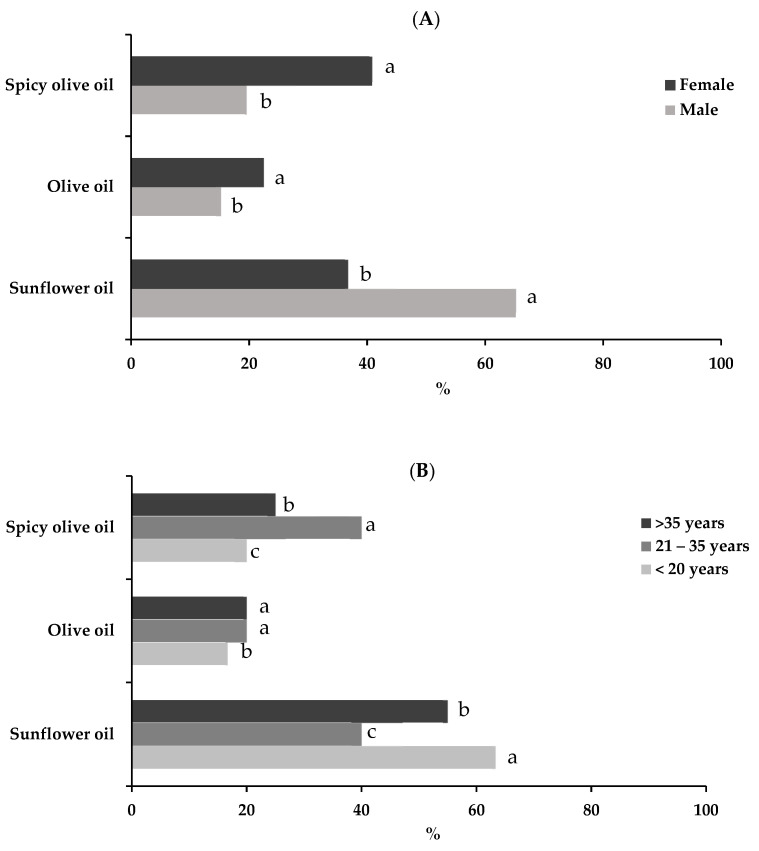
Mean score of preferences by gender (**A**) and age group (**B**). Different letters indicate significant differences (*p* < 0.05) in each type of canned eels.

**Table 1 foods-11-01115-t001:** Descriptive attributes and their definitions adapted from UNE-EN-ISO 5492:2010.

Descriptor	Definition
** *External aspect* **	
***Colour***	Attribute of products inducing a colour sensation. Colour ranging from white to brownish
***Glossiness***	The amount of light reflected from the muscle, ranging from dull to glossy
***Appearance***	All the visible attributes of fish
** *Texture in mouth* **	
***Hardness***	Force required to achieve a given deformation, penetration, or breakage of the fish muscle.
***Adhesiveness***	Force required to remove the muscle with tongue that sticks to the mouth or to a substrate.
***Residual taste***	Sensation perceived whilst the product was in the mouth.
** *Aroma* **	
***Preference***	Preference between two or more samples.
***Intensity***	Perceived strength at differing concentrations of certain volatile compounds
** *Taste* **	
***Bitter***	The intensity of the taste associated with dilute aqueous solutions of various substances such as quinine or caffeine
***Acid***	Taste associated with aqueous solutions of most acid substances such as citric acid.
***Sweet***	Taste associated with aqueous solutions of sugar such as sucrose, or aspartame.
***Salty***	The intensity of the taste associated with salt solutions
***Pungent***	An irritating, sharp sensation, or piercing sensation in the buccal and nasal mucous membranes
***Metallic***	Taste associated with flavour reminiscent of metal, slightly oxidized metal, or salts.
***Aftertaste***	Sensation that occurs after elimination of the product

**Table 2 foods-11-01115-t002:** Colour parameters of raw and canned European eels packed in sunflower oil, olive oil or spiced olive oil at the different steps of the canning process, and after 2 and 12 months of storage. Values are means ± standard deviations of three replicates.

		*L**	*a**	*b**	*H°*	*C**	Δ*E*
**Raw**		74.05 ± 0.24 ^a^	−4.95 ± 0.13 ^ac^	14.60 ± 0.18 ^a^	108.74 ± 0.55 ^a^	15.15 ± 0.35 ^a^	--
**Frying**	**Sunflower oil**	72.68 ± 1.63 ^b^	−5.12 ± 0.20 ^ab^	16.69 ± 0.36 ^c^	108.08 ± 1.67 ^a^	17.56 ± 0.21 ^b^	10.78 ± 0.30 ^ab^
	**Olive oil**	71.94 ± 1.89 ^b^	−4.72 ± 0.51 ^acd^	19.56 ± 0.34 ^de^	103.56 ± 1.18 ^b^	20.13 ± 0.45 ^c^	11.17 ± 0.23 ^a^
**Sterilization**	**Sunflower oil**	77.61 ± 0.70 ^c^	−5.34 ± 0.11 ^b^	18.88 ± 1.07 ^de^	106.02 ± 0.86 ^c^	19.64 ± 1.03 ^c^	11.63 ± 0.06 ^ac^
	**Olive oil**	76.14 ± 1.27 ^c^	−4.68 ± 0.09 ^cd^	21.78 ± 2.95 ^d^	101.84 ± 1.02 ^bd^	22.09 ± 3.22 ^c^	13.31 ± 0.16 ^d^
	**Spiced olive oil**	71.31 ± 1.61 ^b^	−4.92 ± 0.40 ^acd^	19.57 ± 1.48 ^de^	103.74 ± 0.64 ^b^	20.18 ± 1.54 ^c^	11.46 ± 1.38 ^ac^
**2 months storage**	**Sunflower oil**	76.30 ± 1.09 ^c^	−4.69 ± 0.24 ^cd^	19.00 ± 0.56 ^de^	103.88 ± 1.10 ^b^	19.58 ± 0.49 ^c^	10.88 ± 0.23 ^a^
	**Olive oil**	72.76 ± 0.73 ^b^	−4.57 ± 0.15 ^d^	17.61 ± 0.81 ^ce^	104.37 ± 1.21 ^b^	17.59 ± 0.41 ^b^	9.92 ± 0.25 ^b^
	**Spiced olive oil**	67.22 ± 1.58 ^d^	−4.82 ± 0.16 ^c^	18.50 ± 1.24 ^de^	104.08 ± 0.99 ^b^	19.12 ± 1.24 ^c^	12.43 ± 0.51 ^cd^
**12 months storage**	**Sunflower oil**	72.29 ± 1.63 ^b^	−4.35 ± 0.14 ^d^	19.25 ± 0.50 ^de^	102.98 ± 0.23 ^b^	19.75 ± 0.49 ^c^	10.69 ± 0.01 ^ab^
	**Olive oil**	67.22 ± 1.86 ^d^	−3.59 ± 0.21 ^e^	20.02 ± 0.85 ^d^	100.20 ± 0.96 ^d^	20.34 ± 0.81 ^c^	12.31 ± 0.88 ^cd^
	**Spiced olive oil**	70.93 ± 1.75 ^bd^	−4.44 ± 0.04 ^d^	19.25 ± 0.50 ^de^	102.98 ± 0.23 ^b^	19.75 ± 0.49 ^c^	10.70 ± 0.38 ^ab^

^a–e^ Means in the same column with different letters differ significantly (*p* < 0.05).

**Table 3 foods-11-01115-t003:** Colour parameters in the raw oils and after each stage of processing and storage. Values are means ± standard deviations of three replicates.

		*L**	*a**	*b**	*H°*	*C**	Δ*E*
**Raw**	**Sunflower oil**	75.53 ± 2.38 ^ab^	−3.85 ± 0.07 ^a^	4.05 ± 0.06 ^a^	133.50 ± 0.13 ^a^	5.59 ± 0.09 ^a^	--
	**Olive oil**	77.41 ± 1.83 ^b^	−10.49 ± 0.32 ^d^	27.46 ± 1.87 ^b^	110.95 ± 0.73 ^b^	29.40 ± 1.86 ^b^	--
**Frying**	**Sunflower oil**	75.84 ± 3.03 ^ab^	−4.72 ± 0.04 ^b^	7.17 ± 0.64 ^c^	123.46 ± 2.11 ^c^	8.59 ± 0.55 ^c^	4.47 ± 0.35 ^a^
	**Olive oil**	73.08 ± 1.11 ^c^	−9.61 ± 0.06 ^e^	27.78 ± 0.76 ^b^	109.08 ± 0.37 ^b^	29.40 ± 0.73 ^b^	4.48 ± 1.09 ^a^
**Sterilization**	**Sunflower oil**	78.99 ± 1.90 ^bd^	−4.16 ± 0.08 ^c^	4.61 ± 0.46 ^a^	132.66 ± 0.40 ^a^	6.17 ± 0.40 ^a^	4.55 ± 0.63 ^a^
	**Olive oil**	72.97 ± 0.88 ^c^	−9.90 ± 0.36 ^e^	27.04 ± 1.11 ^b^	110.10 ± 0.25 ^b^	29.07 ± 0.92 ^b^	4.53 ± 0.95 ^ab^
	**Spiced olive oil**	72.72 ± 2.13 ^c^	−9.36 ± 0.20 ^ef^	25.62 ± 1.62 ^b^	110.22 ± 0.91 ^b^	27.67 ± 1.46 ^b^	6.08 ± 0.90 ^b^
**2 months storage**	**Sunflower oil**	78.18 ± 1.91 ^bd^	−4.13 ± 0.06 ^c^	4.21 ± 0.43 ^a^	134.58 ± 2.63 ^a^	5.91 ± 1.33 ^a^	4.66 ± 0.97 ^a^
	**Olive oil**	78.10 ± 0.99 ^bd^	−9.96 ± 0.19 ^e^	25.04 ± 0.98 ^b^	111.63 ± 0.41 ^b^	27.04 ± 0.97 ^b^	2.68 ± 0.56 ^c^
	**Spiced olive oil**	82.78 ± 2.13 ^d^	−8.87 ± 0.57 ^f^	19.23 ± 2.66 ^d^	114.63 ± 1.05 ^d^	21.57 ± 1.24 ^d^	10.81 ± 1.02 ^d^
**12 months storage**	**Sunflower oil**	76.79 v ± 1.14 ^b^	−4.89 ± 0.13 ^b^	7.31 ± 0.65 ^c^	124.12 ± 1.73 ^c^	8.65 ± 0.60 ^c^	4.49 ± 0.59 ^a^
	**Olive oil**	76.30 ± 0.99 ^b^	−10.77 ± 0.30 ^d^	32.37 ± 2.27 ^e^	108.46 ± 0.99 ^e^	34.12 ± 2.67 ^e^	2.16 ± 0.72 ^c^
	**Spiced olive oil**	72.89 ± 0.41 ^c^	−10.49 ± 0.17 ^d^	36.25 ± 3.12 ^e^	106.24 ± 1.58 ^e^	37.75 ± 2.95 ^e^	11.87 ± 0.71 ^d^

^a–f^ Means in the same column with different letters differ significantly (*p* < 0.05).

## Data Availability

Data is contained within the article.

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
