# Peer review of "Impact of the Filling Medium on the Colour and Sensory Characteristics of Canned European Eels (Anguilla anguilla L.)"

_foods, 2022, doi:10.3390/foods11081115_

Round 1
Reviewer 1 Report
The manuscript entitled ‘Impact of the filling medium on the colour and sensory characteristics of canned European eels (Anguilla anguilla L.)’ well written and showed very interesting results.
Following some comments:
- Line 97-99, I suggest to make it clear by insert it in a table ‘The hot filling medium was then added….’, also I suggest to use in Treatment 3 instead of ‘olive oil (refined olive oil + virgin olive oil) plus chilli and pepper’ to use spiced olive oil (refined olive oil + virgin olive oil) plus chilli and pepper’ as you used in all manuscript.
- Line 125, 128, Add the full acronym before the abbreviation. ISO and UNE.
- Line 165 please check least squares test (LSD) it should be the least significant difference (LSD) test.
Regards,
Author Response
Reviewer #1:
First of all, we wish to thank the reviewer for his work and dedication to our article. His useful comments and suggestions really contributed to the improvement of our manuscript.
The modifications made in the revised version as a consequence of the Reviewer 1’s suggestions were highlighted with green background
- Line 97-99, I suggest to make it clear by insert it in a table ‘The hot filling medium was then added….’, also I suggest to use in Treatment 3 instead of ‘olive oil (refined olive oil + virgin olive oil) plus chilli and pepper’ to use spiced olive oil (refined olive oil + virgin olive oil) plus chilli and pepper’ as you used in all manuscript.
Following the reviewer’s suggestion, we have changed the phrase. We have also included a diagram to make it clearer.
- Line 125, 128, Add the full acronym before the abbreviation. ISO and UNE.
Following the reviewer’s suggestion, we have added the full acronym
- Line 165 please check least squares test (LSD) it should be the least significant difference (LSD) test.
The reviewer is right. We have changed “least squares test” by “least significant difference test”

Reviewer 2 Report
The article presents a very interesting analysis of a specific product, the canned eels, with emphasis on the oil impact on the sensory and color quality of the canned product. I would recommend some modifications on the article structure in order to improve the information readability.
General topics
Section2.1 Should be better divided.
2.1.1 Samples preparation Line 87-91
2.1.2 cooking and sterilization Line 92-101
2.1.3 Storage Lin 102-107
Section 2.3 – a better description of the sensorial analysis is needed.
I would recommend subsection 2.3.1 for the analysis with the 11 trained evaluators. Subsection 2.3.2 for the consumer test analysis.
Please add the tables used for the sensory analysis, with description of the attributes and the values, not just the description of the attributes, for both the consumer test and the trained evaluators analysis.
Similar approach should be used for the discussion. Also, in the discussion the creation of subtopics for each color parameter evaluated, would be beneficial for an easier read.
Specific
Line 24-26: In my opinion Quality is an overall concept is not an attribute. I recommend the following change:
“Sensory properties are one of the most important quality attribute of any product and directly determines consumer satisfaction or overall acceptance based on colour, taste, flavour and aroma.”
For more information on the definitions of quality is possible to check the other articles.
Line 48-51: Do the authors have any reference or information about the industrial canning process for eel?
Line 96-101: the glass jars where colored? Where the samples stored in dark? The exposure to light wouldn’t have impact on the color evolution of the product and oil oxidation? How the use of glass jar mimics the effect of the commercial cans? Cans don’t allow the interaction with light.
Line 127: why the use of two different standards if the accessor were trained in the ISO standard?
Line 153: 5-6 individuals?
Author Response
First of all, we wish to thank the reviewer for his work and dedication to our article. His useful comments and suggestions really contributed to the improvement of our manuscript.
The modifications made in the revised version as a consequence of the Reviewer 2’s suggestions were highlighted with red background
General topics
Section2.1 Should be better divided.
2.1.1 Samples preparation Line 87-91
2.1.2 cooking and sterilization Line 92-101
2.1.3 Storage Lin 102-107
Following the reviewer’s suggestion, we have divided this section.
Section 2.3 – a better description of the sensorial analysis is needed.
I would recommend subsection 2.3.1 for the analysis with the 11 trained evaluators. Subsection 2.3.2 for the consumer test analysis.
Following the reviewer’s suggestion, we have divided this section.
Please add the tables used for the sensory analysis, with description of the attributes and the values, not just the description of the attributes, for both the consumer test and the trained evaluators analysis.
Following the reviewer’s suggestion, we have added 2 figures with the values
Similar approach should be used for the discussion. Also, in the discussion the creation of subtopics for each color parameter evaluated, would be beneficial for an easier read.
Following the reviewer’s suggestion, we have divided this section.
Specific
Line 24-26: In my opinion Quality is an overall concept is not an attribute. I recommend the following change:
“Sensory properties are one of the most important quality attribute of any product and directly determines consumer satisfaction or overall acceptance based on colour, taste, flavour and aroma.”
For more information on the definitions of quality is possible to check the other articles.
Following the reviewer’s suggestion, we have changed this sentence
Line 48-51: Do the authors have any reference or information about the industrial canning process for eel?
We have not found references on the canning industry for eels.
Line 96-101: the glass jars where colored? Where the samples stored in dark? The exposure to light wouldn’t have impact on the colour evolution of the product and oil oxidation? How the use of glass jar mimics the effect of the commercial cans? Cans don’t allow the interaction with light.
The glass jars were not colored; however, they were stored in dark room. The light can have impact on the colour. This is explicated in discussion.
Line 127: why the use of two different standards if the accessor were trained in the ISO standard?
Sensory analysis was performed according to the criteria of the UNE-ISO 6658 (ISO, 2008) and ISO 11035:1994.
Line 153: 5-6 individuals?
The reviewer is right. We have added “individuals”

Round 2
Reviewer 2 Report
Authors provided the required alterations and answers.
Author Response
We wish to thank the reviewer for his work and dedication to our article.
